# Near-Infrared Spectroscopy-Guided, Individualized Arterial Blood Pressure Management for Carotid Endarterectomy under General Anesthesia: A Randomized, Controlled Trial

**DOI:** 10.3390/jcm12154885

**Published:** 2023-07-25

**Authors:** Tina Tomić Mahečić, Branko Malojčić, Dinko Tonković, Mirabel Mažar, Robert Baronica, Snježana Juren Meaški, Andrea Crkvenac Gregorek, Jens Meier, Martin W. Dünser

**Affiliations:** 1Department of Anesthesiology and Intensive Care Medicine, Clinical Hospital Centre Zagreb, 10000 Zagreb, Croatia; 2Department of Neurology, Clinical Hospital Centre Zagreb, 10000 Zagreb, Croatia; 3Department of Surgery, Clinical Hospital Centre Zagreb, 10000 Zagreb, Croatia; 4Department of Anesthesiology and Intensive Care Medicine, Kepler University Hospital and Johannes Kepler University, 4040 Linz, Austria

**Keywords:** carotid endarterectomy, blood pressure, near-infrared spectroscopy, neurocognitive function, complications

## Abstract

**Background**: Differences in blood pressure can influence the risk of brain ischemia, perioperative complications, and postoperative neurocognitive function in patients undergoing carotid endarterectomy (CEA). **Methods**: In this single-center trial, patients scheduled for CEA under general anesthesia were randomized into an intervention group receiving near-infrared spectroscopy (NIRS)-guided blood pressure management during carotid cross-clamping and a control group receiving standard care. The primary endpoint was postoperative neurocognitive function assessed before surgery, on postoperative days 1 and 7, and eight weeks after surgery. Perioperative complications and cerebral autoregulatory capacity were secondary endpoints. **Results**: Systolic blood pressure (*p* < 0.001) and norepinephrine doses (89 (54–122) vs. 147 (116–242) µg; *p* < 0.001) during carotid cross-clamping were lower in the intervention group. No group differences in postoperative neurocognitive function were observed. The rate of perioperative complications was lower in the intervention group than in the control group (3.3 vs. 26.7%, *p* = 0.03). The breath-holding index did not differ between groups. **Conclusions**: Postoperative neurocognitive function was comparable between CEA patients undergoing general anesthesia in whom arterial blood pressure during carotid cross-clamping was guided using NIRS and subjects receiving standard care. NIRS-guided, individualized arterial blood pressure management resulted in less vasopressor exposition and a lower rate of perioperative complications.

## 1. Introduction

There is firm scientific evidence that carotid stenosis, even when asymptomatic, is associated with progressive neurocognitive decline [1]. Cerebral hypoperfusion and repeated embolisms are suspected as the main causes [2]. Despite of a lack of standardized cognitive tests [3], carotid endarterectomy (CEA) seems to reverse or slow neurocognitive dysfunction in patients with severe carotid artery disease [1]. Among other factors, postoperative recovery of neurocognitive function has been associated with improved cerebral hemodynamics [4].

Inadequate arterial blood pressure control is a key risk factor for the development of brain ischemia, myocardial complications, and perioperative death in patients undergoing CEA [5]. Studies show that increasing arterial blood pressure during carotid cross-clamping can improve cerebral oxygenation [6] and prevent or reverse perioperative neurological deficits [7]. Experts recommend either tolerating arterial hypertension or increasing arterial blood pressure ≥ 20% over baseline values during carotid cross-clamping [5,8,9,10,11]. However, inducing hypertension during carotid cross-clamping may by itself be related to adverse effects, such as inadequate catecholamine exposure, cerebral hyperperfusion injury, increased myocardial oxygen demand, and myocardial ischemia [9,10,12,13]. Despite the abovementioned association of cerebral hemodynamics and postoperative neurocognitive function [4], there are currently no valid data regarding the impact of intraoperative blood pressure management on postoperative neurocognitive function in patients undergoing CEA.

Near-infrared spectroscopy (NIRS) has been introduced as a neuromonitoring tool during carotid artery surgery to track cerebral oxygenation, detect cerebral hypoperfusion [6,14,15], and guide selective shunting [16]. Arterial blood pressure and regional brain tissue oxygenation as measured using NIRS were found to be closely correlated during carotid cross-clamping in patients undergoing CEA [17]. An absolute decrease in regional brain tissue oxygen saturation by 10–20% has been suggested to reflect critical cerebral hypoperfusion [18,19,20].

In this study, we aimed to evaluate whether NIRS-guided arterial blood pressure management during carotid cross-clamping impacts postoperative neurocognitive function, the rate of perioperative complications, and cerebral autoregulatory capacity in patients undergoing CEA. We hypothesized that use of a NIRS-guided management protocol during carotid clamping allows for individualized arterial blood pressure management and results in improved postoperative neurocognitive function, maintained cerebral vascular reactivity, and fewer perioperative complications.

## 2. Materials and Methods

This study was designed as a prospective, investigator-initiated, single-center, randomized, controlled clinical trial. It was conducted at the Department of Anesthesiology and Intensive Care Medicine at the University Hospital Centre, a tertiary, 1800-bed hospital in Zagreb, Croatia. The protocol of this trial was pre-published (www.clinicaltrials.gov, accessed on 1 August 2019; NCT05739357). No changes were made to the study protocol following the commencement of patient enrolment. The study was conducted in accordance with the ethical principles of the Declaration of Helsinki. Ethical approval for this study (Ethical Committee N° 02/21 AG) was provided by the Ethical Committee of Zagreb University Hospital, Zagreb, Croatia (Chairperson Prof. M. Kasum) on 30 January 2018. Written informed consent was obtained from all study patients before study enrolment. The manuscript was prepared according to the updated CONSORT guideline for reporting parallel group randomized trials [21].

### 2.1. Study Patients

Patients scheduled for elective, unilateral CEA because of an internal carotid artery stenosis ≥70% using NASCET criteria [22] were screened for study eligibility during their routine pre-anesthetic evaluation. Patients aged 18–90 years, with American Society of Anesthesiologists classifications of III and higher, and those without pre-existent severe neurocognitive dysfunction (e.g., MoCA > 21 points), were eligible for study inclusion. Patients with a history of recent stroke, those unable to perform the required set of neurocognitive tests (e.g., due to aphasia or hearing or visual impairment), and subjects without an adequate bone window to attain a transcranial Doppler signal were excluded.

### 2.2. Interventions

Study subjects were identified and enrolled by an anesthesiologist, who was part of the research team, during the routine preoperative anesthetic assessment. Study group assignment was then performed in a 1:1 ratio using a computer-generated randomization scheme (Research Randomizer^®^; https://www.randomizer.org/, accessed on 1 January 2019). Subjects allocated to the intervention group were monitored during CEA using bifrontal NIRS (O3^®^; Masimo International, Neuchatel, Switzerland). In the intervention group, arterial blood pressure management during carotid cross-clamping was protocolized and guided using NIRS monitoring (intervention group; see Appendix A). A decrease in the NIRS-derived regional brain tissue oxygen saturation >12% compared to baseline values (measured in the awake patient before general anesthesia induction) indicated norepinephrine titration to increase arterial blood pressure. If increasing arterial blood pressure could not reverse the drop in regional brain tissue oxygen saturation, the anesthesiologist could additionally up-titrate fractional inspiratory oxygen concentration. In subjects allocated to the control group, no NIRS was applied. In these patients, arterial blood pressure during carotid cross-clamping was managed according to institutional standards of care. This standard included titrating norepinephrine to maintain systolic arterial blood pressure within 170–180 mmHg. All anesthetic procedures were delivered by anesthesiologists experienced in vascular anesthesia and formally instructed to implement the NIRS-guided study protocol. The anesthesiologists were not blinded to study group allocation.

All patients included in this study underwent total intravenous anesthesia with propofol and remifentanil (target-controlled infusion). The anesthetic depth was monitored using 4-channel processed electroencephalography (SedLine^®^; Masimo International, Neuchatel, Switzerland; target range: 25–50). During anesthesia, all patients were monitored using V-lead electrocardiography, plethysmographic oxygen saturation, and end-tidal carbon dioxide tension. Arterial blood pressure was measured invasively using an arterial cannula preferentially placed into the radial artery. CEA was performed using the classical technique with patch angioplasty by a group of vascular surgeons according to an institutional protocol. Based on this protocol, a carotid shunt was selectively placed in case of inadequate blood backflow from the distal internal carotid artery in all study patients. The decision to place a carotid shunt was made by the surgeon independent of study group allocation. We deliberately chose not to change the institutional protocol for shunt placement, as our study aimed to specifically determine the effects of arterial blood pressure management during carotid cross-clamping on postoperative neurocognitive function and secondary outcome parameters. At the end of the procedure, a drain was placed in all study patients. Thromboembolic prophylaxis with a low-molecular weight heparin was initiated 12–16 h after the end of surgery. In the morning of postoperative day 1, all patients were put on a single anti-platelet therapy (acetylsalicylic acid, 100 mg q 24 h).

### 2.3. Data Collection

In all study patients, the following data were collected at study inclusion and during the perioperative phase: age, sex, body mass index, dominant hand, years of school education, comorbid conditions, type of antihypertensive drug therapy, American Society of Anesthesiologists physical status classification, presence of neurological symptoms, grade of internal carotid artery stenosis on both sides (as suggested by the North American Symptomatic Carotid Endarterectomy Trial Collaborators) [22], anatomical side of surgery, duration of carotid cross-clamping, need for the placement of a carotid shunt, arterial blood pressure measurements during carotid cross clamping, cumulative dose of norepinephrine administered during carotid cross-clamping, and any perioperative complications. In patients allocated to the intervention group, adherence to the study protocol to manage arterial blood pressure during carotid cross-clamping was evaluated by comparing NIRS data with the anesthetic protocol.

Neurocognitive function was assessed in all study patients at the following time points: before surgery, on the first postoperative day, on postoperative day 7, and eight weeks after surgery. The following neurocognitive tests were conducted according to a standardized, written protocol at the previously mentioned time points: MoCA [23], Trail Making Tests (TMT) A and B [24], and the months backward test (MBT) [25]. All tests were performed by research staff trained in conductance of the respective neurocognitive tests. The Croatian, validated version of the MoCA test was used. Before surgery, on the first postoperative day, and eight weeks after surgery, a breath-holding test was bilaterally performed by a neurologist specialized in neurosonology. Briefly, the mean flow velocity (MFV) of the middle cerebral artery was measured with the use of transcranial Doppler sonography before and at the end of a breath-holding maneuver of at least 30 s [26,27]. The breath-holding index (BHI) was then calculated using the following formula:$$BHI=\frac{100\times \frac{MFV_{end}-MFV_{baseline}}{MFV_{baseline}}}{Breath\;Holding\;Duration}\;$$

Both the examiner conducting the neurocognitive tests and the neurologist performing the breath-holding test were blinded to the group allocation of study patients.

### 2.4. Outcomes

The primary endpoint of this study was postoperative neurocognitive function as assessed using MoCA, TMT, and MBT tests during the eight weeks following CEA surgery. The surgical and non-surgical perioperative complication rates and the postoperative course of the BHI were secondary endpoints. 

### 2.5. Statistical Analysis

This trial was designed to provide a power of 80% to detect a between-group difference in neurocognitive test results at a two-sided significance level of 5%, as assessed by an analysis of variance (assuming four measurements per group). For the trial to have 80% power, it was required to include 30 patients per group with an effect size of 0.25, a correction of repeated measures of 0.5, and a nonsphericity correction of e = 1 (G*Power 3.1.9.6). No interim analysis was performed.

Following double entry of study data into the database and plausibility control of all entered values, statistical analyses were performed by applying the intention-to-treat principle and using the R software package (R version 4.1.2; https://www.R-project.org/, Vienna, Austria, accessed on 9 December 2021). As all datasets for primary and secondary outcome measures were complete, no statistical methods were used to compensate for missing values. Normality assumption of all continuous data was tested using the Kolmogorov–Smirnov test. Descriptive statistical methods were used to report demographic, clinical, and outcome data. Between-group comparisons of primary and secondary endpoints were conducted using the unpaired Student’s *t*-test, Fisher’s exact test, or an analysis of variance for repeated measurements (Tukey test), as appropriate. No subgroup analyses were performed. Data are presented as median values with interquartile ranges or absolute values with percentages, if not otherwise indicated. *p*-values < 0.05 were considered to indicate statistical significance.

## 3. Results

From 1 January 2019 until 23 February 2023, 62 out of 294 patients screened were enrolled in the study. Study enrollment stopped following inclusion of the predetermined sample size. As two patients in the control group were lost to follow-up at eight weeks post-surgery, 32 patients were randomized to the control group. Finally, 60 patients (intervention group, *n* = 30; control group, *n* = 30) were included in the statistical analysis (Figure 1). No cross-over between study groups occurred. No patient included in this study died during, or within eight weeks of, the surgical procedure.

The study groups did not differ in preoperative variables, except there were more patients with higher education levels in the intervention group than in the control group (Table 1). The rate of carotid shunt placements was comparable between the intervention and control groups (7/30 (23.3%) vs. 4/30 (13.3%), respectively; *p* = 0.51). Although the duration of carotid cross-clamping did not differ between the intervention and control groups (37 (32–44) vs. 37 (27–44) min, *p* = 0.98), systolic arterial blood pressure (Figure 2) and the cumulative dose of norepinephrine (89 (54–122) vs. 147 (116–242) µg, *p* < 0.001) during carotid cross-clamping were lower in the intervention group than in the control group.

In all patients allocated to the intervention group, titration of norepinephrine was sufficient to restore regional brain tissue oxygen saturation. No study protocol violations were observed in the intervention group.

### 3.1. Neurocognitive Function

Except for better results in the MBT in the intervention group, postoperative neurocognitive function did not differ between study groups (Figure 3).

Neither relative changes (Figure 4) nor absolute changes in the results of TMT A (−4 (−14–2) vs. −10 (−14–1) s; *p* = 0.75), TMT B (−12 (−51–−2) vs. −25 (−60–8) s; *p* = 0.9), MBT (−5 (−12–7) vs. −2 (−17–5) s; *p* = 0.85) or MoCA (1 (−2–3) vs. 0 (−1–2) points; *p* = 0.63) between the time points before surgery and eight weeks after surgery differed between the intervention and control groups, respectively.

A post-hoc analysis of our test strategy revealed a power of 0.84 for a clinically relevant difference in the MoCA test of 1.0 between groups, and a power of 0.06 for changes over time (R version 4.1.2; https://www.R-project.org/, pwr2 1.0, Vienna, Austria, accessed on 9 December 2021). 

### 3.2. Secondary Outcomes

The rate of perioperative complications was lower in the intervention group compared with the control group (1/30 (3.3%) vs. 8/30 (26.7%); *p* = 0.03) (Table 2). The BHI on the side of surgery or the contralateral side did not differ between study groups during the observation period (Figure 5).

## 4. Discussion

In this prospective, randomized, controlled trial, neurocognitive test results obtained before surgery and at three time points during eight weeks after elective CEA did not differ between patients allocated to a NIRS-guided protocol for arterial blood pressure management during carotid cross-clamping and patients managed using standards of care targeting a systolic arterial blood pressure of 170–180 mmHg. Norepinephrine requirements and the rate of perioperative complications were lower in the intervention group than in the control group. The BHI during the perioperative period did not differ between groups, suggesting that individualized arterial blood management during carotid cross-clamping did not affect the cerebrovascular reserve in this study population [26,27].

Our trial had a >80% post-hoc power to detect a significant effect of the study intervention on postoperative neurocognitive function measured using the MoCA test battery, even if only a one-point difference in MoCA was considered clinically relevant. Taking our results into account, this implies that it was highly likely that NIRS-guided arterial blood pressure management during carotid cross-clamping in study patients undergoing general anesthesia did neither favorably nor adversely affect neurocognitive function within eight weeks after CEA. Although this finding rejects our original hypothesis, the fact that patients allocated to the intervention group experienced fewer perioperative complications suggests that the study intervention could still prove beneficial. As the rate of perioperative complications was only a secondary outcome variable, our trial was not powered to assess this endpoint adequately. However, the fact that study patients managed according to the NIRS-guided protocol had lower arterial blood pressures and were exposed to lower doses of a drug with relevant potential side effects [13,28] is physiologically sound, and could explain the lower rate of perioperative complications observed in the intervention group.

This trial had several strengths. By choosing a randomized, controlled design, an adequate methodology was used to evaluate the effects of the study intervention. The latter proved to be feasible, as highlighted by the absence of protocol violations, and resulted in a clear biological effect indicated by significantly different arterial blood pressures and norepinephrine dosages between the two groups. Although other neuromonitoring tools have been used to detect cerebral hypoperfusion during CEA [14,29,30], we used a NIRS-guided protocol because NIRS is a validated and widely available neuromonitoring technique, easy to use, and can be interpreted without specific training in neurophysiology [6,14,15,16,17,31]. A systematic review and meta-analysis concluded that NIRS has a low sensitivity, but high specificity to identify intraoperative ischemia compared with awake monitoring in patients undergoing CEA [32]. Both postoperative neurocognitive function and the perioperative complication rate were patient-centered study outcomes. Neurocognitive function was assessed with the use of validated assessment tools [23,24,25].

So far, only one study has evaluated the association between arterial blood pressure during carotid cross-clamping and postoperative cognitive function in CEA patients [11]. However, editorial concern has recently been voiced regarding the validity of these data [33]. The effects of intraoperative NIRS monitoring and NIRS-guided arterial blood pressure management on postoperative cognitive function were evaluated in other surgical cohorts. Most of these studies included cardiac surgery patients. A review article summarized there was reasonable agreement that intraoperative decreases in regional cerebral oxygen saturation, as detected using NIRS, were associated with both postoperative delirium and postoperative cognitive decline following cardiac surgery [34]. Preliminary data collected in patients undergoing spinal surgery [35,36], as well as patients during shoulder surgery in the beach chair position [37], similarly suggested that use of an NIRS-based algorithm may help reduce postoperative cognitive disturbances.

On the other hand, certain limitations need to be considered when interpreting the results of our trial. First, strict inclusion and exclusion criteria were applied to identify a homogeneous study population in whom neurocognitive function could be assessed according to established standards. However, this resulted in the exclusion of almost 80% of patients screened for study eligibility and hampers extrapolation of our study results to everyday clinical practice. Second, this was a single-center study, implying that surgical practice (e.g., surgical technique, duration of carotid cross-clamping, shunt placement) and arterial blood pressure management in the control group may vary from those in other institutions. For example, it cannot be excluded that a different arterial blood pressure management protocol during carotid cross-clamping in the control group (e.g., relative increase of the preoperative mean arterial blood pressure >20%) would have resulted in a lower rate of perioperative complications. Third, for obvious reasons, our study could not be double-blinded. Although the researchers conducting the neurocognitive tests were blinded to group allocation, we cannot exclude that a performance or investigator bias influenced the perioperative care of patients randomized to the intervention group. Fourth, the time point of eight weeks after surgery was chosen in line with other studies [38] assuming that patients had sufficient time to recover from the neurocognitive impact of surgery. However, we cannot exclude that testing neurocognitive function at a later time point following CEA [2] might have changed our results.

In conclusion, postoperative neurocognitive function was comparable between CEA patients undergoing general anesthesia in whom arterial blood pressure during carotid cross-clamping was guided by near-infrared spectroscopy and subjects receiving standard care. Near-infrared spectroscopy-guided, individualized arterial blood pressure management resulted in less vasopressor exposition and a lower rate of perioperative complications.

## Figures and Tables

**Figure 1 jcm-12-04885-f001:**
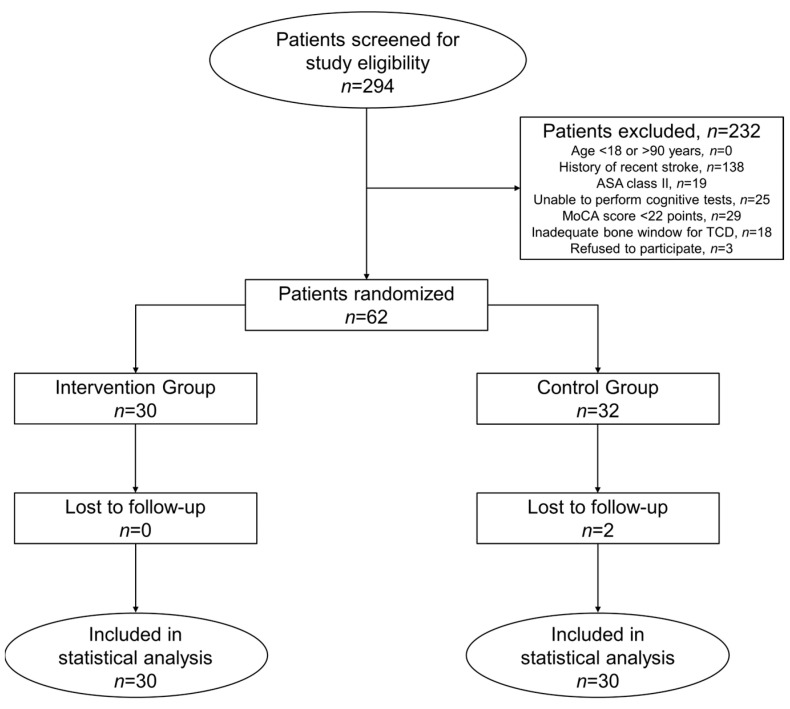
CONSORT flow diagram. ASA, American Society of Anesthesiologists; MoCA, Montreal Cognitive Assessment; TCD, transcranial Doppler sonography.

**Figure 2 jcm-12-04885-f002:**
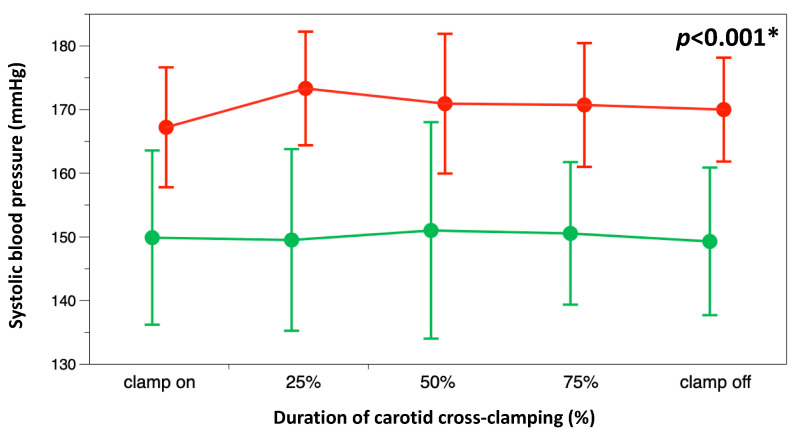
Systolic arterial blood pressures during carotid cross-clamping in both study groups (neuromonitoring or intervention group, green; control group, red). Because of inter-individual differences in the duration of carotid cross-clamping, the duration of carotid cross-clamping is given as a percentage. * *p* < 0.001 for 2-way repeated measurement ANOVA.

**Figure 3 jcm-12-04885-f003:**
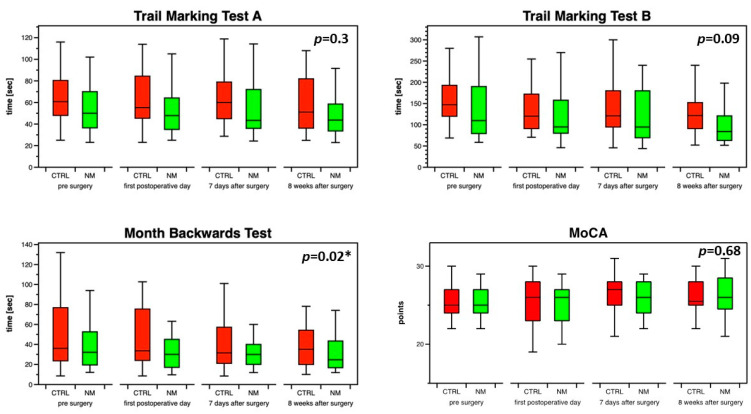
Test results of perioperative neurocognitive function for both study groups. CTRL, control group; MoCA, Montreal Cognitive Assessment; NM, neuromonitoring or intervention group (* *p* < 0.05).

**Figure 4 jcm-12-04885-f004:**
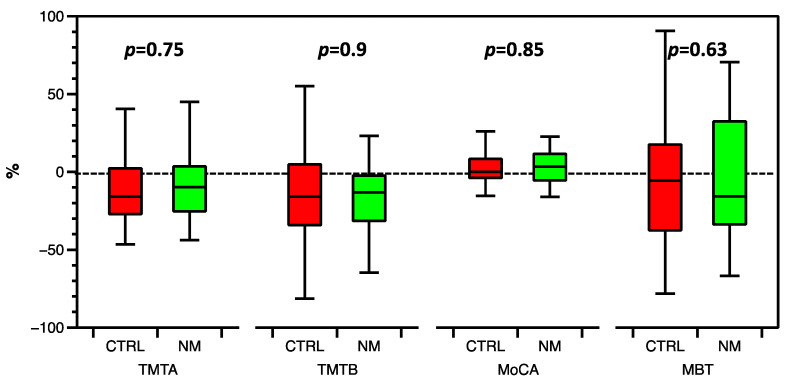
Percentage change in neurocognitive test results at eight weeks after surgery compared with preoperative values for both study groups. CTRL, control group; MBT, months backward test; MoCA, Montreal Cognitive Assessment; NM, neuromonitoring or intervention group; TMTA, Trail Making Test A; TMTB, Trail Making Test B.

**Figure 5 jcm-12-04885-f005:**
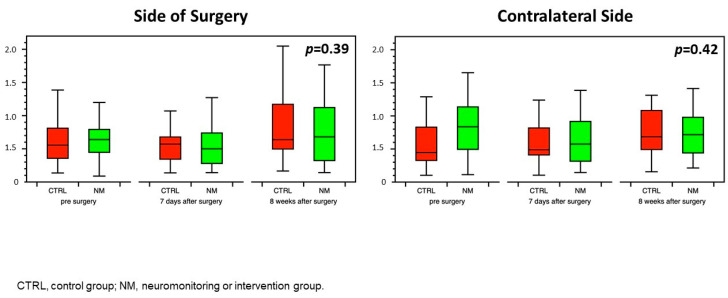
Perioperative course of the breath-holding index on the operated and contralateral sides in both study groups. CTRL, control group; NM, neuromonitoring or intervention group.

**Table 1 jcm-12-04885-t001:** Preoperative characteristics of the study population. ASA, American Society of Anesthesiologists; NASCET, North American Symptomatic Carotid Endarterectomy Trial.

		Intervention Group	Control Group	*p*-Value
n		30	30	
Age	years	70 (63–73)	64 (67–72)	0.76
Female sex	n (%)	5 (17)	9 (30)	0.36
Body Mass Index	kg/m²	27 (26–29)	27 (25–28)	0.6
Right-handedness	n (%)	30 (100)	28 (93.3)	0.49
School education > 12 years	n (%)	19 (63.3)	9 (30)	0.02 *
Comorbid conditions				
chronic arterial hypertension	n (%)	27 (90)	27 (90)	1
congestive heart failure	n (%)	8 (26.7)	9 (30)	1
diabetes mellitus II	n (%)	7 (23.3)	10 (33.3)	0.57
chronic kidney disease	n (%)	1 (3.3)	2 (6.7)	1
chronic obstructive pulmonary disease	n (%)	1 (3.3)	1 (3.3)	1
Antihypertensive drugs				
ACE-inhibitor/sartan	n (%)	17 (56.7)	23 (76.7)	0.17
calcium channel blocker	n (%)	16 (53.3)	15 (50)	1
central antihypertensive	n (%)	10 (33.3)	7 (23.3)	0.57
diuretic	n (%)	9 (30)	16 (53.3)	0.12
beta-blocker	n (%)	9 (30)	14 (46.7)	0.29
alpha-blocker	n (%)	2 (6.7)	1 (3.3)	1
ASA physical status		3 (3–3)	3 (3–3)	0.33
III	n (%)	30 (100)	29 (96.7)	1
IV	n (%)	0 (0)	1 (3.3)	
NASCET stenosis grade				
side of surgery	%	80 (71–90)	85 (80–90)	0.35
contralateral side	%	30 (0–50)	45 (3–54)	0.32
Symptomatic stenosis	n (%)	11 (37)	16 (53.3)	0.3
Montreal Cognitive Assessment test	points	25 (24–27)	25 (24–27)	0.82
Trail marking test A	s	50 (37–69)	61 (48–80)	0.6
Trail marking test B	S	110 (81–186)	147 (120–192)	0.42
Month backward test	s	32 (20–50)	36 (24–75)	0.23
Breath Holding Index				
side of surgery		80 (71–90)	85 (80–90)	0.52
contralateral side		29 (0–50)	45 (3–54)	0.14

ASA, American Society of Anesthesiologists; NASCET, North American Symptomatic Carotid Endarterectomy Trial. *, significant difference between the intervention and control groups. Data are given as median values with interquartile ranges, if not otherwise indicated.

**Table 2 jcm-12-04885-t002:** Perioperative complications in the intervention and control groups.

Intervention Group				Control Group			
*n* = 30				*n* = 30			
Type of Complication	*n* (%)	Postop Day	Management	Type of Complication	*n* (%)	Postop Day	Management
Arterial hypertension *	1 (3.3%)			Rebleeding	3 (10%)		
	Patient 1	2	Antihypertensive		Patient 1	1	surgical revision
			therapy				
					Patient 2	1	surgical revision
					Patient 3	1	surgical revision
				Arterial hypertension *	3 (10%)		
					Patient 1	2	antihypertensive therapy
					Patient 2	3	antihypertensive therapy
					Patient 3	2	antihypertensive therapy
				Unstable angina	1 (3.3%)		
					Patient 1	2	anti-ischemic, antihypertensive therapy
				New neurological deficit	1 (3.3%)		
					Patient 1	0	symptomatic care as per stroke protocol

*, persistent arterial blood pressure readings >200/100 mmHg.

## Data Availability

Data are unavailable due to privacy and ethical restrictions.

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
