# Peer review of "Near-Infrared Spectroscopy-Guided, Individualized Arterial Blood Pressure Management for Carotid Endarterectomy under General Anesthesia: A Randomized, Controlled Trial"

_jcm, 2023, doi:10.3390/jcm12154885_

Round 1
Reviewer 1 Report
The present manuscript is trying to shed light on the role of the individualized blood pressure (BP) control for patients undergoing carotid endarterectomy (CEA) under general anaesthesia, using near-infra-red spectroscopy (NIRS) for monitoring and managing of blood pressure changes.
It is a randomized controlled trial between NIRS guided BP management versus standard of care concluding that: the cognitive functions were similar while the perioperative complications and the vasopressor exposure were less after NIRS guided management.
The abstract and the title are informative about the manuscript aim and results. “The Introduction” is well-written and informative about what is already known about the subject. The aim, research question as well as the study end-points are clear.
Comments:
1. Shunt use was apparently selective, as depicted from results and tables, however, the criteria to use shunt selectively were not clearly mentioned in section 2.2. (Interventions).
2. The authors mention that blood pressure and inspiratory oxygen concentration up-titration were the only measures taken for 12% decrease in brain tissue oxygen saturation from the baseline. Why there was no role for selective shunts in this case?
3. In results: perioperative complications need to be linked with a possible reason for each; rebleeding, chest pain. What was the time of occurrence of each complication, and what was the management required.
4. Was the blood pressure over 180 during operations in the control group or more vasopressor responsible for some perioperative complications such as arterial hypertension, re-bleeding, chest pain, or the new neurological deficit?
5. Is the post-operative protocol entailed using single or double anti-platelet or using a prophylactic anti-coagulant or not?
6. The discussion lacks comparison of the study findings with previous studies.
Author Response
Reviewer 1:
- Shunt use was apparently selective, as depicted from results and tables, however, the criteria to use shunt selectively were not clearly mentioned in section 2.2. (Interventions).
Authors’ Response: The following statement was added to the Methods section of the revised manuscript: “CEA was performed using the classical technique with patch angioplasty by a group of vascular surgeons according to an institutional protocol. Based on this protocol, a carotid shunt was selectively placed in case of inadequate blood backflow from the distal internal carotid artery in all study patients. The decision to place a carotid shunt was made by the surgeon independent of study group allocation.”
- The authors mention that blood pressure and inspiratory oxygen concentration up-titration were the only measures taken for 12% decrease in brain tissue oxygen saturation from the baseline. Why there was no role for selective shunts in this case?
Authors’ Response: We thank the reviewer for this relevant question. As given above, the decision to place a carotid shunt was made by the vascular surgeon independent of study group allocation. We deliberately chose not to change the institutional protocol for shunt placement as our study aimed to specifically determine the effects of arterial blood pressure management during carotid cross-clamping on postoperative neurocognitive function and secondary outcome parameters. This information has been added to the Methods section of the revised manuscript as well.
In addition, the following statement has already been included as a limitation of the study paragraph of our manuscript: “Second, it was a single-center study implying that surgical practice (e.g., surgical technique, duration of carotid cross-clamping, shunt placement) and arterial blood pressure management in the control group may vary from those in other institutions.”
- In results: perioperative complications need to be linked with a possible reason for each; rebleeding, chest pain. What was the time of occurrence of each complication, and what was the management required.
Authors’ Response: In the revised manuscript, we have expanded Table 2 and added information on when (during the postoperative period) the complication occurred and what its management was. However, we had difficulties to objectively define causes for each of the complications. Like in clinical practice, for most complications it was difficult to identify one specific cause and therefore establish causality. Although each complication, particularly rebleeding, arterial hypertension and chest pain, could well be explained by induced arterial hypertension, we refrained from mentioning postulated causes of each complication.
Please note that we have corrected a mistake which has occurred during drafting of Table 2 in the original version. The lines for the number of patients experiencing arterial hypertension and a new neurological deficit were exchanged. This has been corrected in the revised version. We apologize for this inaccuracy and mistake in the original submission.
- Was the blood pressure over 180 during operations in the control group or more vasopressor responsible for some perioperative complications such as arterial hypertension, re-bleeding, chest pain, or the new neurological deficit?
Authors’ Response: The following statement has already been included in the Discussion section of our manuscript: “As the rate of perioperative complications was only a secondary outcome variable, our trial was not powered to assess this endpoint adequately. However, the fact that study patients managed according to the NIRS-guided protocol had lower arterial blood pressures and were exposed to lower doses of a drug with relevant potential side effects [9,25] is physiologically sound and could explain the lower rate of perioperative complications as observed in the intervention group.”
Please also refer to our response to this reviewer’s third comment.
- Is the post-operative protocol entailed using single or double anti-platelet or using a prophylactic anti-coagulant or not?
Authors’ Response: The following statement was added to the Methods section of the revised manuscript: “Thromboembolic prophylaxis with a low-molecular weight heparin was initiated 12-16 hours after the end of surgery. In the morning of postoperative day 1, all patients were put on a single anti-platelet therapy (acetylsalicylic acid, 100 mg q24 hrs).”
- The discussion lacks comparison of the study findings with previous studies.
Authors’ Response: In the Discussion section of the revised manuscript, we compared the results of our trial to those reported by authors of other studies. As this is the first study specifically evaluating neurocognitive outcomes associated with neuromonitoring-guided arterial blood pressure control during carotid cross-clamping in CEA patients, we compared our results to those of studies evaluating the association between intraoperative NIRS monitoring and postoperative cognitive function in other surgical populations. Accordingly, the following new paragraph was added to the Discussion section of the revised manuscript: “So far, only one study has evaluated the association between arterial blood pressure during carotid cross-clamping and postoperative cognitive function in CEA patients [4]. However, editorial concern has recently been voiced regarding the validity of these data [doi:10.1227/neu.0000000000002565]. The effects of intraoperative NIRS monitoring and NIRS-guided arterial blood pressure management on postoperative cognitive function were evaluated in other surgical cohorts. Most of these studies included cardiac surgery patients. A review article summarized that there was reasonable agreement that intraoperative decreases in regional cerebral oxygen saturation, as detected by NIRS, were associated with both postoperative delirium and postoperative cognitive decline following cardiac surgery [doi:10.1053./j.jvca.2021.07.029]. Preliminary data collected in patients undergoing spinal surgery [doi:10.1016/j.ijsu.2015.02.009; doi:10.3390/medicina55050179] as well as patients during shoulder surgery in the beach chair position [doi:10.1097/CORR.0000000000001864] similarly suggested that use of a NIRS-based algorithm may help to reduce postoperative cognitive disturbances.”
Reviewer 2 Report
1. The Authors should discuss results of their trial in the context of other studies - it is a missing part of the Discussion chapter.
2. Minor. This paper is written in American English, still "ischaemia" (line 44) is in British English - should be corrected
This paper is written in American English, still "ischaemia" (line 44) is in British English - should be corrected
Author Response
Reviewer 2:
- The Authors should discuss results of their trial in the context of other studies - it is a missing part of the Discussion chapter.
Authors’ Response: In the Discussion section of the revised manuscript, we compared the results of our trial to those reported by authors of other studies. As this is the first study specifically evaluating neurocognitive outcomes associated with neuromonitoring-guided arterial blood pressure control during carotid cross-clamping in CEA patients, we compared our results to those of studies evaluating the association between intraoperative NIRS monitoring and postoperative cognitive function in other surgical populations. Accordingly, the following new paragraph was added to the Discussion section of the revised manuscript: “So far, only one study has evaluated the association between arterial blood pressure during carotid cross-clamping and postoperative cognitive function in CEA patients [4]. However, editorial concern has recently been voiced regarding the validity of these data [doi:10.1227/neu.0000000000002565]. The effects of intraoperative NIRS monitoring and NIRS-guided arterial blood pressure management on postoperative cognitive function were evaluated in other surgical cohorts. Most of these studies included cardiac surgery patients. A review article summarized that there was reasonable agreement that intraoperative decreases in regional cerebral oxygen saturation, as detected by NIRS, were associated with both postoperative delirium and postoperative cognitive decline following cardiac surgery [doi:10.1053./j.jvca.2021.07.029]. Preliminary data collected in patients undergoing spinal surgery [doi:10.1016/j.ijsu.2015.02.009; doi:10.3390/medicina55050179] as well as patients during shoulder surgery in the beach chair position [doi:10.1097/CORR.0000000000001864] similarly suggested that use of a NIRS-based algorithm may help to reduce postoperative cognitive disturbances.”
- Minor. This paper is written in American English, still "ischaemia" (line 44) is in British English - should be corrected
Authors’ Response: Throughout the revised manuscript, “ischaemia” was corrected to “ischemia”.
Reviewer 3 Report
Dear Authors,
I would like to thank you for your interesting study. Your primary and secondary endpoints are clearly pointed in the text. Your statistical analysis has been properly made. There are some issues that must be analyzed in your study.
In Table 1 you don't refer the medical history of the patients ( arterial hypertension, heart failure, renal failure, medication of patients). These are very important because cerebral autoregulation is totally different in patients with AH compared to normotensive patient. Additionally specific chronic antihypertensive treatment like calcium channel blockers alters cerebral autoregulation in a group of patients.Please improve table 1 patients characteristics.
In table 2 you mentioned the perioperative complications. Could you explain why the number of complications are higher in control group? All procedures performed by one vascular surgeon?
What do you mean when you write "chest pain"? Did the patient had any cardiac ischemic event or troponin rise? Please refer.
Did you put a drainage in all patients undergoing carotid CEA,please refer.
What was the antiplatelet therapy in both groups postoperatively? Please refer it in the text.
Your target systolic hypertension during carotid clamping is 170-180mmHg. My opinion is that it is not appropriate for all patients underwent CEA. It would be better to use MAP>20% as a more accurate method to augment pressure during clamping. I believe that it is quite high increasing perioperative events after declamping.
In discussion you haven't mentioned any studies evaluating NIRS in carotid surgery. Please in order to increase citations, please provide a literature review
I would like to thank you for one more time for your study and i am waiting for your replies.
Author Response
Reviewer 3:
- In Table 1 you don't refer the medical history of the patients (arterial hypertension, heart failure, renal failure, medication of patients). These are very important because cerebral autoregulation is totally different in patients with AH compared to normotensive patient. Additionally specific chronic antihypertensive treatment like calcium channel blockers alters cerebral autoregulation in a group of patients. Please improve table 1 patients characteristics.
Authors’ Response: In the revised manuscript, Table 1 includes and compares information on comorbid conditions as well as the type of antihypertensive drugs as suggested by this reviewer. We found no significant differences in these parameters between the two study groups. According changes were also made to the Methods section of the revised manuscript.
- In table 2 you mentioned the perioperative complications. Could you explain why the number of complications are higher in control group? All procedures performed by one vascular surgeon?
Authors’ Response: Regarding the number of vascular surgeons who performed carotid endarterectomy in this study population, the following statement was added to the Methods section of the revised manuscript: “CEA was performed using the classical technique with patch angioplasty by a group of vascular surgeons according to an institutional protocol.”
In the revised manuscript, we have expanded Table 2 and added information on when (during the postoperative period) the complication occurred and what its management was. However, we had difficulties to objectively define causes for each of the complications. As in clinical practice, for most complications it was difficult to identify one single cause and therefore establish causality. Although each complication, particularly rebleeding, arterial hypertension and chest pain, could well be explained by induced arterial hypertension, we refrained from mentioning postulated causes of each complication.
The following statement has already been included in the Discussion section of our manuscript: “As the rate of perioperative complications was only a secondary outcome variable, our trial was not powered to assess this endpoint adequately. However, the fact that study patients managed according to the NIRS-guided protocol had lower arterial blood pressures and were exposed to lower doses of a drug with relevant potential side effects [9,25] is physiologically sound and could explain the lower rate of perioperative complications as observed in the intervention group.”
Please note that we have corrected a mistake which has occurred during drafting of Table 2 in the original version. The lines for the number of patients experiencing arterial hypertension and a new neurological deficit were exchanged. This has been corrected in the revised version. We apologize for this inaccuracy and mistake in the original submission.
- What do you mean when you write "chest pain"? Did the patient had any cardiac ischemic event or troponin rise? Please refer.
Authors’ Response: Thank you very much for raising this point. In addition to the abovementioned changes made to Table 2, we have specified chest pain in the patient mentioned as “unstable angina”. This patient who did not have a history of previous anginal symptoms, experienced chest pain on postoperative day 2. Chest pain was accompanied by ST-segment depressions in the ECG but no elevations in troponin levels. Accordingly, a diagnosis of unstable angina was established by the consulting cardiologist. Treatment consisted of anti-anginal (beta-blocker) and antihypertensive (calcium-channel blocker) therapy. According changes were made to Table 2 in the revised manuscript.
- Did you put a drainage in all patients undergoing carotid CEA,please refer.
Authors’ Response: The following statement was added to the Methods section of the revised manuscript: “At the end of the procedure, a drainage was placed in all study patients.”
- What was the antiplatelet therapy in both groups postoperatively? Please refer it in the text.
Authors’ Response: The following statement was added to the Methods section of the revised manuscript: “Thromboembolic prophylaxis with a low-molecular weight heparin was initiated 12-16 hours after the end of surgery. In the morning of postoperative day 1, all patients were put on a single anti-platelet therapy (acetylsalicylic acid, 100 mg q24 hrs).”
- Your target systolic hypertension during carotid clamping is 170-180mmHg. My opinion is that it is not appropriate for all patients underwent CEA. It would be better to use MAP>20% as a more accurate method to augment pressure during clamping. I believe that it is quite high increasing perioperative events after declamping.
Authors’ Response: The reviewer raises a valid point. It is conceivable that different results might have been observed if the arterial blood pressure in control patients had been managed using a target of a 20% relative increase in mean arterial blood pressure. Accordingly, we have mentioned this limitation in the Discussion section of the revised manuscript. The following statement was added: “For example, it cannot be excluded that a different arterial blood pressure management protocol during carotid cross-clamping in the control group (e.g., relative increase of the preoperative mean arterial blood pressure by >20%) would have resulted in a lower rate of perioperative complications.”
- In discussion you haven't mentioned any studies evaluating NIRS in carotid surgery. Please in order to increase citations, please provide a literature review
Authors’ Response: In the Discussion section of the revised manuscript, we compared the results of our trial to those reported by authors of other studies. As this is the first study specifically evaluating neurocognitive outcomes associated with neuromonitoring-guided arterial blood pressure control during carotid cross-clamping in CEA patients, we compared our results to those of studies evaluating the association between intraoperative NIRS monitoring and postoperative cognitive function in other surgical populations. Accordingly, the following new paragraph was added to the Discussion section of the revised manuscript: “So far, only one study has evaluated the association between arterial blood pressure during carotid cross-clamping and postoperative cognitive function in CEA patients [4]. However, editorial concern has recently been voiced regarding the validity of these data [doi:10.1227/neu.0000000000002565]. The effects of intraoperative NIRS monitoring and NIRS-guided arterial blood pressure management on postoperative cognitive function were evaluated in other surgical cohorts. Most of these studies included cardiac surgery patients. A review article summarized that there was reasonable agreement that intraoperative decreases in regional cerebral oxygen saturation, as detected by NIRS, were associated with both postoperative delirium and postoperative cognitive decline following cardiac surgery [doi:10.1053./j.jvca.2021.07.029]. Preliminary data collected in patients undergoing spinal surgery [doi:10.1016/j.ijsu.2015.02.009; doi:10.3390/medicina55050179] as well as patients during shoulder surgery in the beach chair position [doi:10.1097/CORR.0000000000001864] similarly suggested that use of a NIRS-based algorithm may help to reduce postoperative cognitive disturbances.”
Furthermore, we have added a statement on the use of intraoperative NIRS monitoring in patients undergoing CEA. The respective part of the revised Discussion section reads as follows: “Although other neuromonitoring tools have been used to detect cerebral hypoperfusion during CEA [11,26,27], we used a NIRS-guided protocol because NIRS is a validated and widely available neuromonitoring technique, easy to use and can be interpreted without specific training in neurophysiology [2,10–14]. A systematic review and meta-analysis concluded that NIRS has a low sensitivity but high specificity to identify intraoperative ischemia compared with awake monitoring in patients under-going CEA [doi:10.1016/j.ejvs.2021.08.022].”
- I would like to thank you for one more time for your study and i am waiting for your replies.
Authors’ Response: We thank you for your comments and efforts to improve our manuscript!
Round 2
Reviewer 2 Report
Accept in present form
Author Response
Thank you very much for your revision.
Reviewer 3 Report
Dear Authors,
You have properly made all corrections in your text and you answered all of my questions. Congratulations for your work
Please in line42 (Introduction) you mean cerebral hypoperfusion and not hyperperfusion. Please correct
Author Response
Thank you very much for pointing out this typo. We changed it.